# Comparing Endovascular Treatment Methods in Acute Ischemic Stroke Due to Tandem Occlusion Focusing on Clinical Aspects

**DOI:** 10.3390/life11050458

**Published:** 2021-05-20

**Authors:** Peter Janos Kalmar, Gabor Tarkanyi, Csaba Balazs Nagy, Peter Csecsei, Gabor Lenzser, Edit Bosnyak, Zsofia Nozomi Karadi, Adam Annus, Istvan Szegedi, Andras Buki, Laszlo Szapary

**Affiliations:** 1Department of Neurology, Medical School, University of Pécs, 7623 Pécs, Hungary; kalmar.peterj@gmail.com (P.J.K.); tarkanyigabor@indamail.hu (G.T.); bosnyak.edit@pte.hu (E.B.); karadi.zsofia@pte.hu (Z.N.K.); 2Department of Neurosurgery, Medical School, University of Pécs, 7623 Pécs, Hungary; nagy.csaba2@pte.hu (C.B.N.); csecseipeti@yahoo.com (P.C.); lenzser.gabor@pte.hu (G.L.); buki.andras@pte.hu (A.B.); 3Department of Neurology, University of Szeged, 6725 Szeged, Hungary; annus.adam@med.u-szeged.hu; 4Department of Neurology, University of Debrecen, 4032 Debrecen, Hungary; szegedi.istvan@med.unideb.hu

**Keywords:** stroke, tandem occlusion, endovascular treatment, thrombectomy, acute stenting

## Abstract

Introduction: Acute ischemic strokes (AIS) due to tandem occlusion (TO) of intracranial anterior large vessel and concomitant extracranial internal carotid artery (EICA) are represent in 15–20% of all ischemic strokes. The endovascular treatment (EVT) strategy for those patients is still unclear. Although the intracranial mechanical thrombectomy (MT) is considered as a standard treatment approach, the EICA lesion stent necessity remains a matter of debate. We sought to assess the efficacy and safety of EVT in tandem lesions, particularly the EICA stenting management. Methods: We retrospectively analyzed all patients with anterior circulation stroke associated with EICA lesion and receiving EVT in the three participated stroke centers between November 2017 and December 2020. Patients’ data were collected from our prospective stroke registry (STAY ALIVE). Patients enrolled in our study were divided into two groups depending on whether acute carotid stenting (ACS) or balloon angioplasty only (BAO) technique was used. Our primary outcome was the 90-day functional outcome assessed by modified Rankin scale (mRS). Mortality at 90 days and symptomatic intracranial hemorrhage (sICH) were considered as secondary outcomes. Results: A total of 101 patients (age: 67 ± 10 years, 38.6% female) were enrolled in our study, including 29 (28.3%) BAO cases, and 72 (71.3%) patients treated with ACS. Patients in the BAO group were slightly older (70 ± 9 years vs. 66 ± 10 years, *p* = 0.054), and had higher prevalence of comorbidities such as hypertension (100.0% vs. 59.4%, *p* < 0.001). There was no significant difference in favorable outcomes (51.7% vs. 54.4%, *p* = 0.808) between the groups. However, we observed a trend towards higher rates of sICH (8.3% vs. 3.4%, *p* = 0.382) and 90-day mortality (23.5% vs. 13.8%, *p* = 0.278) with significantly higher frequency of distal embolization (39.1% vs. 17.9%, *p* = 0.043) in patients with ACS. In the overall population age (*p* = 0.013), atrial fibrillation (AF) (*p* = 0.008), National Institutes of Health Stroke Scale (NIHSS) baseline (*p* = 0.029), and successful recanalization (*p* = 0.023) were associated with favorable outcome. Conclusion: Endovascular approach of EICA in addition to MT was safe and effective in tandem occlusion of anterior circulation. Furthermore, our results suggest that balloon angioplasty technique without acute stenting shows a comparable favorable outcome rate to ACS with moderately less hemorrhagic events and mortality rates.

## 1. Introduction

Acute ischemic stroke (AIS) due to tandem occlusion (TO) is defined as high grade stenosis or occlusion of the cervical segment of the internal carotid artery (ICA) associated with concurrent ipsilateral intracranial large vessel occlusion (LVO) along the anterior cerebral circulation (primarily in the distal ICA or middle cerebral artery (MCA) segments) [1,2]. This type of AIS accounts for 15–20% of all ischemic strokes [3]. Approximately 40–69% of these patients survive with severe neurological deficits or die without effective treatment [4].

Recent randomized-controlled trials (RCT) have shown the efficacy and safety of endovascular treatment (EVT) of LVO in the anterior circulation [5,6,7]. However, in these trials, patients with TO were usually excluded (SWIFT PRIME, EXTEND-IA) or poorly represented (18.3% in REVASCAT, 17% in ESCAPE) due to the greater stroke severity and complex endovascular technique of TO [5,7,8,9,10]. Consequently, endovascular treatment (EVT) of patients with TO is mostly described in small and retrospective single-center studies, therefore, there is a lack of strong evidence for the optimal management of this subtype. Current guidelines recommend performing mechanical thrombectomy (MT) if it is reasonable and administering intravenous thrombolysis (IVT) in every eligible patient [11]. However, IVT has moderate recanalization success in tandem lesions (achieves recanalization in only 9–22%), depending on the site of occlusion [12,13,14]. Furthermore, currently there is no consensus on the ideal technical interventional strategy. Multiple endovascular treatment strategies have been described: balloon angioplasty with or without emergent carotid artery stenting of the extracranial ICA (EICA) preceded or followed by MT [15]. However, the appropriate patient population for the different endovascular strategies are still unclear [16,17,18,19].

The aim of our study was to compare the efficacy and safety of acute carotid stenting (ACS) and balloon angioplasty only (BAO) techniques in patients with AIS due to atherothrombotic anterior tandem occlusion.

## 2. Methods

### 2.1. Study Population

This multicenter-study was based on three major stroke institutions’ prospective clinical registry (STAY ALIVE Acute Stroke Registry). Research protocol was approved by the local ethics committee. For all patients enrolled in the study, written consent was given in accordance with Good Clinical Practice (GCP) guidelines.

All consecutive patients with acute stroke due to TO and have received EVT between November 2017 and December 2020 were screened and retrospectively analyzed. TO was defined as an occlusion or high-grade stenosis (NASCET (North American Symptomatic Carotid Endarterectomy Trial) >70%) of EICA with a concomitant ipsilateral intracranial LVO (terminal segment of ICA (tICA), MCA M1 or M2 branch) [20].

Patients meeting the following criteria were included: (1) age over 18 years; (2) acute stroke symptoms with baseline National Institutes of Health Stroke Scale (NIHSS) >4 or isolated aphasia or hemianopia was recognized; (3) symptom onset was no longer than 24 h; (4) TO confirmed by either CT angiography, peri-interventional angiograms, or both; and (5) acute EVT with ACS or BAO of EICA and intracranial MT. Cases of primary ACS or BAO without intracranial MT or occlusion due to EICA dissection were excluded. Furthermore, datasets including inconsistent information were also excluded.

We collected the following data: demographic variables; vascular risk factors; medication therapy; stroke severity (National Institutes of Health Stroke Scale (NIHSS) score); baseline clinical and labor parameters; admission and control imaging data (Alberta Stroke Program Early CT Score (ASPECTS); multiphase CT-angiography (mCTA) collateral score; time metrics; interventional techniques features and complications.

### 2.2. Diagnosis

Patients either arrived primarily in the emergency department of the major neurovascular centers (primary transport) or within the hospitals of surrounding cities and were transferred to the major stroke centers (secondary transport).

Baseline NIHSS was evaluated by a dedicated stroke neurologist on admission. Every patient underwent CT angiography under the center’s protocol for AIS, to assess the intracranial LVO and EICA lesion, and to exclude intracranial hemorrhage. If TO was suspected from the initial CT image, it was confirmed with digital subtraction angiography (DSA).

### 2.3. Thrombolysis

IVT was administered (0.9 mg/kg) in every eligible patient within a maximum 4.5 h after stroke onset (clinical and laboratory inclusion and exclusion criteria for IVT were applied).

### 2.4. Endovascular Technique

EVT was performed under conscious sedation or general anesthesia. This decision was made by the neurointerventional specialist and the anesthesiologist on call. DSA was performed with transfemoral approach; if this was not feasible, we chose radial approach. Biplane angiography was used for the intervention. After the local anesthesia in the inguinal region, an 8F guiding or balloon-guiding catheter (Neuronmax, Penumbra, Alameda, CA, USA; Guider Softip, Boston Scientific, Marlborough, MA, USA; Flowgate 2, Stryker, Kalamazoo, MI, USA) was introduced into the common carotid artery and angiography was performed to evaluate the EICA lesion. The applied endovascular technique was left to the discretion of the physician. In general, the anterograde approach (proximal-to-distal) was applied. A microwire (Roadrunner, Cook Medical, Bloomington, IN, USA) was navigated through the EICA occlusion up to the petrosus segment. At this point the carotid artery stenting (Roadsaver, Boston Scientific, Marlborough, MA, USA; Wallstent, Boston Scientific, Marlborough, MA, USA) with balloon angioplasty (Aviator, Cordis, Santa Clara, CA, USA; Sterling, Boston Scientific, Marlborough, MA, USA) was performed under flow arrest, after this maneuver the aspiration catheter was navigated into the level of LVO and the intracranial MT was performed. MT was performed with a direct aspiration catheter (SOFIA, Microvention, Aliso Viejo, CA, USA) or in combination with a stent-retriever (SOFIA; Solitaire-FR, EV3, Irvine, CA, USA) if it was necessary.

The BAO technique was used primarily in high-risk patients with increased hemorrhagic diathesis, because of the unnecessity of periprocedural antiplatelet therapy and less traumatic effect for the vessels. In these patients a transform balloon (Aviator, Cordis, Santa Clara, CA, USA; Sterling, Boston Scientific, USA) was placed into the catheter to the level of the ICA lesion and was inflated submaximally. The catheter with the inflated balloon gently traversed through the affected segment. The atherosclerotic wall depositions of ICA were removed with manual aspiration, following which, thrombectomy was performed.

The thrombolysis in cerebral infarction (TICI) score was assessed on digital subtraction angiography at the end of the procedure. Successful recanalization was defined as TICI 2b or 3 [21].

### 2.5. Antiplatelet Regimen

During the procedure, a 500 mg bolus of aspirin IV was administered directly prior to acute stenting. No periprocedural anti-aggregation was given in the BAO group. After the procedure, every patient received a preventive dosage of low-molecular-weight heparin (LMWH) to prevent deep vein thrombosis and pulmonary embolism. After the exclusion of the hemorrhagic transformation on the 24-h control image, dual antiplatelet therapy with oral acetylsalicylic acid at 100 mg and clopidogrel at 75 mg daily was initiated in the stenting group and continued for 3 months. As secondary prevention, BAO patients received anti-platelet monotherapy with oral acetylsalicylic acid at 100 mg lifelong [22].

### 2.6. Follow-Up

Early neurologic improvement was assessed based on the 24-h and 72-h NIHSS score after admission. A control CT scan was performed to evaluate the infarction volume and hemorrhagic status 24 h after the procedure. The hemorrhagic transformations were classified according to the European Cooperative Acute Stroke Study (ECASS II) classification [23]. Symptomatic intracranial hemorrhage (sICH) was defined as parenchymal hematoma (PH1 and PH2) with increase of 4 points on the NIHSS. The carotid stent patency was examined with duplex ultrasound by the departments’ stroke neurologists 24 h after the procedure.

The primary outcome was the modified Rankin scale (mRS) score at 90 days with scores ranging from 0 (no symptoms) to 6 (death), mRS 0–2 being considered a good functional outcome. Safety endpoints were sICH and 90-day all cause-mortality.

### 2.7. Statistical Analysis

Data analysis was performed using SPSS (version 26.0, IBM, New York). The Kolmogorov–Smirnov test was used to test normality. Data is presented as mean ± SD or median and interquartile range, where appropriate. Categorical data were compared using the Χ^2^ or the Fisher exact test. Student t test or Mann–Whitney U test were used for the comparison of continuous variables. Binary logistic regression analysis was used to assess the association between baseline data and outcomes. Adjustment was made for potential confounders, variables with *p* < 0.1 in the univariate analysis were entered into the multivariable logistic regression model. A *p* value of <0.05 was considered statistically significant.

## 3. Results

Altogether we enrolled 101 patients with acute stroke due to tandem occlusion that was treated by endovascular approach between November 2017 and December 2020 in the three participating institutions. EICA was treated with ACS in 72 (71.3%) and BAO in 29 (28.7%) patients. The characteristics of the overall population is summarized in Table 1 and subgroups based on EICA treating method in Table 2.

The mean age (±SD) was 67 ± 10 years and patients were predominantly male (61.4%). Forty-nine patients (48.5%) arrived by secondary transport to the neurovascular centers. Patients with BAO were slightly older (66 ± 10 vs. 70 ± 9, *p* = 0.054), and had higher prevalence of comorbidities such as hypertension (100.0% vs. 59.4%, *p* < 0.001). Intracranial occlusion sites were tICA in 30 patients (29.7%), MCA M1 in 55 patients (54.5%) and MCA M2 in 16 patients (15.8%). The median baseline NIHSS was 12 (IQR 9–16) with median ASPECTS 9 (IQR 8–9) and median mCTA 4 (IQR 3–4) without significant differences. Prior IVT before thrombectomy was administered in 27 patients (26.7%).

Regarding the details of intervention, the median symptom onset to arterial puncture time was 347 (IQR 230–655) minutes. The procedure time was slightly shorter in the BAO group (43, IQR 30–60 vs. 49, IQR 33–65, *p* = 0.450), however the difference was not significant.

Successful recanalization was achieved in 83.2% of the patients and was more frequent in the ACS group (86.1% vs. 75.9%, *p* = 0.213), furthermore the number of complications during the procedure (9.7% vs. 3.4%, *p* = 0.291) and incidence of distal embolization (39.1% vs. 17.9%, *p* = 0.043) showed an upward trend towards for acute stent treatment. Symptomatic intracranial hemorrhage (sICH) was detected in 7 patients (6.9%), however, there was a slightly higher occurrence of bleeding events, aICH (20.0% vs. 10.7%, *p* = 0.273) and sICH (8.3% vs. 3.4%, *p* = 0.382) in the ACS group.

In the 90-day follow-up 52 patients (53.6%) showed favorable outcome and 20 patients (20.6%) died. Four patients (4.0%) were lost to follow-up. In the ACS group, 37 (54.4%) patients achieved good functional outcome and 15 (51.7%) in the BAO group. Mortality rate at 90 days was slightly lower in patients with BAO (13.8% vs. 23.5%, *p* = 0.278). There were no significant differences in the outcomes.

Regression analysis showed that younger age (*p* = 0.013), lack of atrial fibrillation (AF) (*p* = 0.008), low baseline NIHSS score (*p* = 0.029), and successful recanalization (*p* = 0.023) were associated with favorable outcome, while older age (*p* = 0.046), alcohol consumption (*p* = 0.033), high NIHSS score at 72 h (*p* = 0.009), low mCTA score (*p* = 0.039), and presence of sICH (*p* = 0.034) were associated with 90-day mortality. Distal embolization did not prove to be an independent predictor of the outcomes. Independent predictors of outcomes are summarized in Table 3.

Furthermore, we performed a subgroup analysis to compare the sICH and non-sICH patient groups, results are summarized in Appendix A. Besides the 90-day mortality, the rate of intraprocedural complications and primary transports were higher in the sICH patient group. Although these results should be interpreted cautiously due to the low number of sICH cases.

## 4. Discussion

In this multicenter analysis, patients treated with the angioplasty only technique achieved comparable favorable outcome rates to the acute stent group. However, we observed a trend towards higher rates of hemorrhages and 90-day mortality, with significantly higher frequency of distal embolization in patients with acute stenting.

Recent randomized-controlled trials have shown the superiority of EVT in LVOs over IVT alone [5,6,7]. However, approximately 15–20% of patients with AIS have high grade stenosis or occlusion of EICA in addition to intracranial LVO [3]. This type of acute stroke has been poorly represented in RCTs, therefore data from RCTs are limited. Consequently, there is no consensus on the optimal EVT technique of tandem lesions. Currently, single-center studies have presented that acute stenting of EICA is technically feasible with high success rate. However, the rate of hemorrhage events ranged from 18 to 43% and stent thrombosis in up to 17% of cases have been described [24]. Because of the aforementioned details, multiple EVT approaches have been reported, although the majority of studies deal primarily in more depth with the approach of vascular openings (proximal-to-distal or distal-to-proximal) and there are few studies and lack of guidance about the EICA opening management.

In our present study 52 (53.6%) patients achieved favorable outcome at 90 days, 20 patients died (20.6%), and sICH occurred 7 (6.9%) times. Our results are consistent with the findings of previous studies and confirm the efficacy and necessity of EVT in tandem lesions [19]. However, these studies were mainly single-center studies with small numbers of BAO patients and heterogeneous periprocedural methods, furthermore, all of these studies primarily applied stent-retriever technique for MT during the procedure, whereas we used the direct aspiration technique predominantly [19]. In 2020, Xing et al. directly investigated the two MT techniques efficacy in tICA occlusions. Patients treated with the direct aspiration technique achieved significantly higher successful recanalization rates which may explain the slightly better successful recanalization rate in our study compared to previous studies (83.6% vs. 78%) [19,25].

Furthermore, the rate of favorable outcome was slightly higher in the ACS patients’ group (54.4% vs. 51.7%), however, at the same time, a moderately higher proportion of hemorrhagic events (aICH, 20.0% vs. 10.7%; sICH, 8.3% vs. 3.4%) and 90-day mortality (23.5% vs. 13.8%) were observed. The intravenous antiplatelet therapy directly prior to carotid stenting may be the explanation of the relatively higher hemorrhage events, however, it is necessary to prevent acute stent thrombosis. Currently only few data are available for periprocedural antiplatelet management of acute carotid stenting during EVT [22]. Based on the latest antiplatelet Delphi consensus, aspirin IV (500 mg bolus) should be used as the first-line agent prior to carotid stenting [22]. Although Da Ros et al. identified potential predictors for sICH as the higher intraprocedural heparin dosage, the initial ASPECTS ≤7 and the MT needs more than one attempt for complete recanalization [26]. However, differences were not observed in our study, probably due to the significantly lower number of sICH cases, the different periprocedural heparin management, and the lack of detailed MT attempts data (only the presence of first pass effect was documented).

In our study, 20% of patients had AF and 29 patients received antithrombotic medication at admission. Therefore, our study’s population had an increased hemorrhagic diathesis, which was further increased by the prior IVT (26.7%). Besides the neurointerventionalist’s individual discretion, every patient’s individual bleeding risk was also assessed and considered before choosing the EVT strategy.

ACS patients with AF are recommended postprocedural triple inhibition based on expert opinion, however, it has been proven to raise the bleeding rates [27,28]. Our results suggest that these bleeding events could probably be avoided by treating EICA with BAO.

Data are also limited in terms of stent patency at the subacute phase of patients with TO. The mechanism of thrombus formation related to mutual interaction between the coagulation system and platelet activator system is due to vascular endothelial damage and individual patient-, lesion-, and stent-related factors (poor stent expansion, delayed endothelization, inflammation reactions) [27,29]. Five (8.8%) subacute stent thrombosis were detected by 24-h carotid ultrasound in our study. This stent thrombosis rate is similar that reported by Wallocha et al. (5.6%) [30].

Only few previous studies investigated the rate of distal embolization according to the interventional technique. Li et al. reported that they found higher distal embolic rate in the angioplasty only group [31]. However, occurrence of distal embolization in our study was significantly higher in patients with ACS (39.1% vs. 17.9%, *p* = 0.043). The difference can be explained with our PTA technique in the BAO group, which was very similar to the balloon assisted technique (BAT) in cardiology, therefore, after the submaximally inflated balloon was gently traversed to the entire affected segment, then manual aspiration of atherosclerotic deposits was performed [32,33].

The main strength of our study is the thorough investigation of EVT strategies in TO including the treatment technique with direct aspiration catheter and with the unique balloon angioplasty technique. The evaluation of the rate of distal embolization according to the EVT technique should also be highlighted as it had not been observed in previous larger studies. However, our study has several limitations. Firstly, this is a nonrandomized retrospective analysis with a relatively small number of patients. Secondly, the heterogeneity of treating methods due to different endovascular approaches between centers may have affected the outcomes. Finally, due to retrospective registry design we cannot rule out the existence of potentially important variables that were not included in our analysis.

## 5. Conclusions

In conclusion, an endovascular approach of EICA in addition to MT was safe and effective in tandem occlusion of anterior circulation, regardless of the opening technique. However, our results suggest that the balloon angioplasty technique without acute stenting showed a comparably favorable outcome rate as a contrast to ACS with moderately less hemorrhagic events and mortality rates, therefore, BAO may be a suitable alternative treatment for patients with high bleeding risk. A prospective randomized study would be warranted to specifically clarify the best treatment strategies for patient groups.

## Figures and Tables

**Table 1 life-11-00458-t001:** Demographic variables and clinical parameters of overall patient population.

	TO Patients (N = 101)
Age, years, mean (±SD)	67 (±10)
Gender, female, % (*n*)	38.6 (39)
Smoking, % (*n*)	66.2 (43)
Alcohol, % (*n*)	38.5 (25)
Hypertension, % (*n*)	71.4 (70)
Diabetes mellitus, % (*n*)	20.6 (20)
Dyslipidemia, % (*n*)	45.4 (44)
Atrial fibrillation, % (*n*)	19.8 (20)
Previous stroke % (*n*)	18.1 (17)
Antiplatelet inhibitor therapy at admission, % (*n*)	22.8 (21)
Aspirin, % (*n*)	47.6 (10)
Clopidogrel, % (*n*)	33.3 (7)
Dual antiplatelet (Aspirin + Clopidogrel), % (*n*)	19.0 (4)
Oral anticoagulant therapy at admission, % (*n*)	8.7 (8)
NIHSS baseline, median (IQR)	12 (9–16)
NIHSS 24h, median (IQR)	8 (4–13)
NIHSS 72h, median (IQR)	7 (4–11)
Early ischemic sign on admission CT, % (*n*)	70.7 (70)
ASPECTS, median (IQR)	9 (8–9)
mCTA score, median (IQR)	4 (3–4)
LVO site	
tICA, % (*n*)	29.7 (30)
MCA M1, % (*n*)	54.5 (55)
MCA M2, % (*n*)	15.8 (16)
Contralateral EICA stenosis, % (*n*)	22.7 (22)
Primary transport, % (*n*)	51.5 (52)
Symptom to arterial puncture time median (IQR)	347 (230–655)
Puncture to revascularization time, median (IQR)	47 (33–64)
Symptom onset to revascularization, median (IQR)	400 (275–725)
IVT prior MT, % (*n*)	26.7 (27)
First pass effect, % (*n*)	53.6 (52)
TICI ≥ 2b, % (*n*)	83.2 (84)
Aspiration catheter, % (*n*)	79.2 (80)
Combined MT, % (*n*)	20.8 (21)
Complications, % (*n*)	7.9 (8)
None, % (*n*)	92.1 (93)
Dissection, % (*n*)	3.0 (3)
Perforation, % (*n*)	1.0 (1)
SAH, % (*n*)	3.0 (3)
Other, % (*n*)	1.0 (1)
Distal embolization, % (*n*)	33.0 (32)
aICH, % (*n*)	17.3 (17)
sICH, % (*n*)	6.9 (7)
90-day mRS ≤ 2, % (*n*)	53.6 (52)
90-day mortality, % (*n*)	20.6 (20)

Abbreviations: TO, tandem occlusion; SD, standard deviation; IQR, interquartile range; NIHSS, National Institutes of Health Stroke Scale; ASPECTS, Alberta Stroke Program Early CT Score; mCTA, multiphase CT-angiography; LVO, large vessel occlusion; EICA, extracranial internal carotid artery; IVT, intravenous thrombolysis; TICI, thrombolysis in cerebral infarction; MT, mechanical thrombectomy; SAH, subarachnoidal hemorrhage; sICH, symptomatic intracranial hemorrhage; aICH asymptomatic intracranial hemorrhage; mRS, modified Rankin scale.

**Table 2 life-11-00458-t002:** Evaluated parameters in acute carotid stenting (ACS) and balloon angioplasty only (BAO) patient groups.

	ACS (N = 72)	BAO (N = 29)	*p*
Age, years, mean (±SD)	66 ± 10	70 ± 9	0.054
Gender, female, % (*n*)	34.7 (25)	48.3 (14)	0.206
Smoking, % (*n*)	66.0 (31)	66.7 (12)	0.957
Alcohol, % (*n*)	38.3 (18)	38.9 (7)	0.965
Hypertension, % (*n*)	59.4 (41)	100.0 (29)	**<0.001**
Diabetes mellitus, % (*n*)	20.3 (14)	21.4 (6)	0.900
Atrial fibrillation, % (*n*)	16.7 (12)	27.6 (8)	0.213
Dyslipidemia, % (*n*)	42.6 (29)	51.7 (15)	0.411
Previous stroke % (*n*)	19.4 (13)	14.8 (4)	0.601
API therapy at admission, % (*n*)	18.2 (12)	34.6 (9)	0.091
OAC therapy at admission, % (*n*)	6.1 (4)	15.4 (4)	0.153
NIHSS baseline, median (IQR)	12 (9–16)	13 (9–16)	0.450
NIHSS 24h, median (IQR)	8 (4–13)	8 (6–11)	0.591
NIHSS 72h, median (IQR)	7 (4–10)	8 (6–11)	0.432
ASPECTS, median (IQR)	8 (8–9)	9 (8–9)	0.213
mCTA score, median (IQR)	4 (3–4)	4 (3–4)	0.938
First pass effect, % (*n*)	55.1 (38)	50.0 (14)	0.650
TICI ≥ 2b, % (*n*)	86.1 (62)	75.9 (22)	0.213
Symptom onset to arterial puncture time median (IQR)	360 (235–655)	310 (215–665)	0.838
Puncture to revascularization time, median (IQR)	49 (33–65)	43 (30–60)	0.450
Symptom onset to revascularization, median (IQR)	400 (275–725)	385 (270–680)	0.832
Complications, % (*n*),	9.7 (7)	3.4 (1)	0.291
Distal embolization, % (*n*)	39.1 (27)	17.9 (5)	**0.043**
aICH, % (*n*)	20.0 (14)	10.7 (3)	0.273
sICH, % (*n*)	8.3 (6)	3.4 (1)	0.382
Early stent thrombosis, % (*n*)	8.8 (5)	-	
90-day mRS ≤ 2, % (*n*)	54.4 (37)	51.7 (15)	0.808
90-day mortality, % (*n*)	23.5 (16)	13.8 (4)	0.278

Abbreviations: ACS, acute carotid stenting; BAO, balloon angioplasty only; SD, standard deviation; IQR, interquartile range; API, antiplatelet inhibitor; OAC, oral anticoagulant; NIHSS, National Institutes of Health Stroke Scale; ASPECTS, Alberta Stroke Program Early CT Score; mCTA, multiphase CT-angiography; IVT, intravenous thrombolysis; TICI, thrombolysis in cerebral infarction; sICH, symptomatic intracranial hemorrhage; aICH asymptomatic intracranial hemorrhage; mRS, modified Rankin scale.

**Table 3 life-11-00458-t003:** Independent predictors of outcomes.

	OR (95% CI)	*p*
90-day mRS 0–2		
Age	0.932 (0.882–0.985)	0.013
Atrial fibrillation	0.142 (0.034–0.600)	0.008
NIHSS baseline	0.901 (0.821–0.989)	0.029
Successful recanalization (TICI ≥ 2b)	5.653 (1.271–25.145)	0.023
90-day mortality		
Age	1.132 (1.002–1.279)	0.046
Alcohol consumption	13.356 (1.239–143.935)	0.033
NIHSS 72h	1.248 (1.056–1.476)	0.009
mCTA score	0.521 (0.281–0.966)	0.039
sICH	15.264 (1.228–189.710)	0.034

Abbreviations: OR, odds ratio; CI, confidence interval; mRS, modified Rankin scale; NIHSS, National Institutes of Health Stroke Scale; TICI, thrombolysis in cerebral infarction; mCTA, multiphase CT-angiography; sICH, symptomatic intracranial hemorrhage.

## Data Availability

The data presented in this study are available on request from the corresponding author. The data are not publicly available due to patient privacy considerations (HIPPA).

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
