# Peer review of "Comparing Endovascular Treatment Methods in Acute Ischemic Stroke Due to Tandem Occlusion Focusing on Clinical Aspects"

_life, 2021, doi:10.3390/life11050458_

Round 1
Reviewer 1 Report
This is generally a well-written and comprehensive article comparing the efficacy and safety of acute carotid stenting and balloon angioplasty only technique, in patients with acute ischemic stroke due to atherothrombotic anterior tandem occlusion. The study is correctly designed and the results are clearly and transparently presented. The set goals correspond to the conclusions. Even though this study is retrospective and included a relatively small number of patients, I consider that the findings are interesting. However, given that there are other recent studies that are very similar to this one, I recommend to highlight the original elements of the study.
Author Response
Dear Reviewer,
Many thanks for you for the time and effort made to review our manuscript and to provide thorough and insightful comments on our work. Based on your review we could significantly improve the manuscript. Our point-by-point responses for the comments are presented in the attachment. The changes in the manuscript are highlighted in red. We hope that we sufficiently answered the comments and the changes made will satisfy you.
Yours sincerely,
Laszlo Szapary MD, PhD
Corresponding Author

Reviewer 2 Report
This was a retrospective study aimed to assess the efficacy and safety of EVT in stroke patients with tandem occlusions according to treatment with acute carotid stenting (ACS) or balloon angioplasty. Balloon-angioplasty technique was associated with a comparable outcome and a moderately lower rate of haemorrhagic events and mortality than ACS.
The study is interesting, and results are nicely presented. There are, however, some issues that need to be further addressed.
During carotid stent placement and mechanical thrombectomy for tandem occlusion treatment, higher intraprocedural heparin dosage (≥3000 IU) has been associated with increased symptomatic intracranial hemorrhage risk when the initial ASPECTS was ≤7, and mechanical thrombectomy needed more than one passage for complete recanalization. It would be appropriate to put the study findings into the current research context and mention which are the variables that have been identified as risk factor for symptomatic intracranial hemorrhage after ACS (Ref. Carotid Stenting and Mechanical Thrombectomy in Patients with Acute Ischemic Stroke and Tandem Occlusions: Antithrombotic Treatment and Functional Outcome. Am J Neuroradiol 2020).
Author Response

(The authors gave the same response as above.)

Round 2
Reviewer 2 Report
The Authors addressed all the issues.